# Trends in Seaweed Extract Based Biostimulants: Manufacturing Process and Beneficial Effect on Soil-Plant Systems

**DOI:** 10.3390/plants9030359

**Published:** 2020-03-12

**Authors:** Mohammed EL Mehdi EL Boukhari, Mustapha Barakate, Youness Bouhia, Karim Lyamlouli

**Affiliations:** 1AgroBioSciences Program, Mohammed 6 Polytechnic University UM6P, Benguerir 43150, Morocco; Mohamed.ELBOUKHARI@um6p.ma (M.E.M.E.B.); Mustapha.BARAKATE@um6p.ma (M.B.); Youness.BOUHIA@um6p.ma (Y.B.); 2Faculty of Sciences Semlalia, Laboratory of Microbial Biotechnology, AgroSciences and Environment, Cadi Ayyad University, Marrakesh 40000, Morocco

**Keywords:** seaweed extract, biostimulant, mechanism of action, abiotic stress, regulation, nutrient use efficiency

## Abstract

The time when plant biostimulants were considered as “snake oil” is erstwhile and the skepticism regarding their agricultural benefits has significantly faded, as solid scientific evidences of their positive effects are continuously provided. Currently plant biostimulants are considered as a full-fledged class of agri-inputs and highly attractive business opportunity for major actors of the agroindustry. As the dominant category of the biostimulant segment, seaweed extracts were key in this growing renown. They are widely known as substances with the function of mitigating abiotic stress and enhancing plant productivity. Seaweed extracts are derived from the extraction of several macroalgae species, which depending on the extraction methodology lead to the production of complex mixtures of biologically active compounds. Consequently, plant responses are often inconsistent, and precisely deciphering the involved mechanism of action remains highly intricate. Recently, scientists all over the world have been interested to exploring hidden mechanism of action of these resources through the employment of multidisciplinary and high-throughput approaches, combining plant physiology, molecular biology, agronomy, and multi-omics techniques. The aim of this review is to provide fresh insights into the concept of seaweed extract (SE), through addressing the subject in newfangled standpoints based on current scientific knowledge, and taking into consideration both academic and industrial claims in concomitance with market’s requirements. The crucial extraction process as well as the effect of such products on nutrient uptake and their role in abiotic and biotic stress tolerance are scrutinized with emphasizing the involved mechanisms at the metabolic and genetic level. Additionally, some often overlooked and indirect effects of seaweed extracts, such as their influence on plant microbiome are discussed. Finally, the plausible impact of the recently approved plant biostimulant regulation on seaweed extract industry is addressed.

## 1. Introduction

In the last fifty years, the agroindustry has embodied major achievements by taking advantage of the chemical revolution, which enabled the emergence of mineral fertilizers. This later fruitful alliance of both the agronomical and chemical sciences was a critical driver of the dramatic improvement that yield of most agricultural crops have known, thus adapting to the nutritional surge generated by demographic boom post-World War II. Currently, in a context where environmental pressures due to global warming are pervasive, modern agriculture is assuredly facing which might be its greatest challenge, as nutritional needs of a world population expecting to reach 9.7 billion by 2050 [1], are maintaining the same exponential trend, concurrently with sizeable reduction of arable lands and soil degradation. According to a United Nations report, in Europe 970 million tons of fertile soils have been getting lost each year, and approximately 24 billion tons worldwide due to erosions and inadequate agricultural practices [2]. Furthermore, the quandary develops into severely convoluted conjunctures, when bearing in mind marked drought issues, increased salinization of cultivated lands [3] and the impact of more frequent natural disasters, which for example was responsible between 2003 and 2013 of a loss of 80 billion USD in terms of crops and livestock production [4].

Today to counterbalance the abovementioned deleterious effects, novel approaches should be considered in developing innovative agro-solutions aiming at establishing functional and efficacious sustainable intensive agriculture systems. Such initiatives are peculiarly needed in fast-growing regions population-wise, like sub-Saharan Africa and where agriculture, in addition to being broadly dominated by archaic practices, is tremendously affected by high uncertainty due to climate variability [5]. That being said, the agroindustry has hardly remained placid in the face of aforementioned problems, and solution are continuously provided to answer requirements of small holders and large crop producers. Development of smart irrigation systems aiming at optimizing water usage, water-soluble fertilizers, sets of good agricultural practices, biocontrol agents and, more lately, promising slow- and controlled-release fertilizers [6], are all examples of solid innovations which contributed to improving agriculture by several folds. Moreover, the recent introduction of precision agriculture tools into farming practices like remote sensing through advanced drones equipped with high-tech imaging devices (i.e., RGB camera, flurocam, hyperspectral imaging, etc.) and the recent introduction of the big data concept [7], holds out the prospect of significant growth in terms of fertilization and crop management.

In this pool of innovations, extracts derived from seaweed are standing as a novel class of agro inputs originating from the horticultural world, which have particularly attracted the interest of both industrial and scientific communities and continue to do so. Field and greenhouse evidences of their benefits for crops’ production are continuously provided in profuse manners. In fact, the use of seaweeds in agriculture dates back thousands of years ago. During the ancient roman times, plant seedlings were covered with algae to promote their growth. In the coastal area of Europe, farmers incorporated seaweeds in the soil or used it as a compost [8]. Starting from 1948, 18 countries had developed their seaweed resources for fertilizers; and by 1947, Milton succeeded in making a liquid product, which could be considered as the founding stone of the seaweed extract industry with respect to agricultural applications [9].

Currently seaweed extracts (SE) are widely used as plant biostimulants, which are ‘any substance or microorganism applied to plants with the aim to enhance nutrition efficiency, abiotic stress tolerance and/or crop quality traits, regardless of its nutrients content’ [10]. Seaweed extracts constitutes more than 33% of the total biostimulant market worldwide and are predicted to reach a value of 894 million Euro in 2022 [11]. Moreover, it is estimated that seaweeds or macroalgae compromise nearly 10,000 species [12], which are subdivided mainly to three categories based on their pigmentation, Phaeophyta (Brown), Rhodophyta (Red), and Chlorophyta (Green) [13]. Brown seaweeds with *Ascophyllum*, *Fucus*, *Laminaria* are the dominant group [10].

Seaweeds extracts biochemical composition is complex (polysaccharides, minerals, vitamins, oils, fats, acids, antioxidants, pigments, hormones) [9,13,14]. Hence understanding their mechanism of action is highly intricate, and often requires multidisciplinary approach due the multiple interaction between the substantial numbers of bioactive compound within the same extract. SE can be applied on soil and/or on plants as a foliar spray [10]. They act positively on soil retention and remediation, and soil microflora, they could be a source of nutrients, and they may show hormonal effects [10].

After more than a decade of commercialization and scientific data production, the question arises, where does SE stands currently? Therefore, the aim of this review is to provide fresh insights into the concept of SE, through addressing the subject in newfangled standpoints based on current scientific knowledge and taking into consideration both academic and industrial claims in concomitance with market’s requirements. An in-depth critical analysis of the highly determining extraction process will be carried out. With regards to agrophysiological aspects, the effect of such products on nutrient uptake and their role in abiotic stress tolerance will be scrutinized with emphasizing the involved mechanisms at the metabolic and genetic level. Furthermore, the standing of SE regulation in the context of the new European Fertilizing Products Regulation (FPR) will be discussed.

## 2. Manufacturing Process of Seaweed Extract Biostimulant: The Most Critical Challenge in SE Development

In SE manufacturing process, the extraction phase remains largely the most decisive step for the development of agronomically efficient products. Unlike the extraction of polysaccharide or pharmaceutical substances, which target a specific compound, the complexity is much higher as the extraction process aim to guarantee the integrity of a maximum of biologically active molecules with presumed biostimulant effect. Such logical approach is not always optimal, as several aspects could be overlooked, namely, the targeted crop, the targeted physiological effect and most importantly the phenological characteristic of algae species with respect to seasonal variation. Bearing in mind this assumption it is not surprising that, for the majority of manufacturers, the extraction process is the key aspect that gives them a competitive advantage. Therefore, these internally-developed methodologies, which are generally based on soft extractions (low pressure, low temperature), are often the subject of professional secrecy. As a general rule, the objective of any given extraction technology is to reach targeted compounds with a high yield, low energy consumption, by-products preservation, reduced waste production, and optimized extraction process [15]. Currently different methods are used to this purpose. Novel extraction technologies, like ultrasound-assisted extraction (UAE), enzyme-assisted extraction (EAE), supercritical fluid extraction (SFU), microwave-assisted extraction (MAE), and pressurized liquid extraction (PLE) offer the advantage of extracting biological compounds without affecting their activity [16]. However, classical extraction technologies are thought to be more solvents and time consuming [17]. More importantly, depending on the extraction process, the yield of bioactive compounds fluctuates significantly. For instance, extract obtained through water or acid hydrolysis are reported to be rich in phytohormones [18], although such trait may also be inconsistent. Indeed, the previous findings of Ertani et al. [19] revealed notable variation of indole acetic acid (IAA) and gibberellic acid (GA) content of five commercial *Ascophyllum nodosum*-based biostimulants produced via acid extraction, which was plausibly attributed to differences in the sampling zone, timing, and environmental conditions. At the industrial level, the most widely used extraction process implies heating the algae biomass with potassium hydroxide or sodium solvents in pressurized reaction vessels [9]. As a result, compounds with a biostimulant potential could be lost during this process [20]. Additionally, in alkali extraction, polysaccharide chains are broken down to smaller oligomers, and new compounds not initially present in the algae biomass could be formed [18], which could be advantageous or not as the specific chemical process is not fully controlled. To resolve such issues, the use of UAE could constitute a potential solution. For example, UAE enabled the reduction of carrageenans and alginates extraction time from (*Kappaphycus alvarezii* and *Euchema denticulatum*) and (*Sargassum binderi* and *Turbinaria ornata*), respectively, without altering their chemical structure and molar mass distribution [21]. Moreover, the extraction of high molecular weight phenolic compounds could be further enhanced [22]. Ultimately, even if those extraction techniques have been improved for decades, which led to efficient biostimulant products, the level of variability is still significant as to respond to industrial requirements. In that regard enzyme-assisted extraction (EAE) could be one of the most promising technologies to achieve extraction consistency as it could be easily tailored to target specific compounds depending on the characteristic of algae biomass. EAE is a green technology consisting in the application of cell wall degrading enzymes to macroalgae at optimal conditions of pH and temperature to take advantage of the targeted bioactive compounds [17]. A wide range of enzymes is used in this process as pepsin, cellulase, viscozyme, alcalse, and flavourzyme [23,24,25]. In a recent study, Vasquez et al. [26] showed that the application of cellulase to *Macrocystis pyrifera* and *Chondracanthus chamissoi* resulted in protein extraction yields of 76.6% and 36.1%, respectively. However, the extraction time consumption exceeded 12 h. On the other hand, Okolie et al. [27] compared several methods of extraction of fucoidan from *Ascophyllum nodosum* and showed that EAE produced lesser yield than the conventional extraction using chemicals. Yet, fucose and galactose contents were identical. Overall, recently major advances have been achieved in SE manufacturing process, however the consistency challenge is still to be tackled. The extraction technique should be designed with respect to the biomass features, the mode of application, the crop’s type, and the desired physiological effect (e.g., abiotic stress tolerance, improving soil fertility, improving fruit quality, etc.), rather than aiming to extract a maximum of bioactive compounds, as various interactions may occur within the mixture leading to both synergistic and antagonistic effects. Finally, integrated extraction approaches (Figure 1), which make use of combined techniques, should be highly considered as to exploit the full algae biomass.

## 3. Overview on the Positive Effect of Seaweed Extracts on Plant Productivity

Seaweed extracts have been used widely in agriculture during the last recent years to promote crops productivity. Such amelioration is obtained through stimulation of different physiological processes involved in plant growth and development, as well as improvement of last product quality (Figure 2). For instance, Trivedi et al. [28] investigated the effect of the application of *Kappaphycus alvarezii* seaweed extract on maize at grain filling stage and reported an increase of mainly 15% of the seed yield (g/plant) in water optimal conditions through the enhancement of yield parameters as the number of seeds per cob and the cob length. Different application rates of *Kappaphycus alvarezii* were investigated in respect to 100% of the recommended dose of fertilizers on maize [29] and potato tubers [30], interestingly, 7.5% was found to be the best application rate at both studies. Similarly, Mattner et al. [31] applied a commercial seaweed extract based on *Duvillaea potatorum* and *Ascophyllum nodosum* on strawberry at nursery and production stages, results showed an increase of the root length density of plants which had been treated at the production stage regardless of whether the treatment was applied at the nursery stage or not. This increase was highly correlated with the enhancement of the final yield recorded. Consequently, they suggested that seaweed extracts could be involved in enhancing plant water and nutrients use efficiency, and supposed that other seaweed components different from minerals, like phytohormones might be involved in this process. In this context, different studies were conducted to investigate seaweed extracts as a source of phytohormones. For instance, Kulkarni et al. [32] studied the phytohormone like activity of different biostimulants in spinach, and revealed that cytokinins cis-zeatin, dihydrozeatin, and isopentyladenine significantly were increased in plants treated with a pure eckol compound isolated from *Ecklonia maxima*. Cytokinins are supposed to play an important role in plant growth regulation [33].

A pre-plantation treatment of kelpack, a commercial product based on *Ecklonia maxima* seaweed, on willow plants was able to enhance the growth of the stem for both cultivars Inger and Tordis, and a mixture of kelpack and triacontanol enhanced the shoot surface area [34]. Furthermore, Kelpack treatment enhanced the electron transfer rates (ETRs) for both photosystems I and II of Tordis cultivar. Similarly, Al-Ghamdi et al. [35] reported a significant increase of the total chlorophylls content, photosynthetic rate, transpiration rate, and stomatal conductance of asparagus plants when treated by a 7 mL.L^−1^
*Ascophyllum nodosum* seaweed extract in comparison to the untreated plants. This enhancement was marked by an increase of the branch length and dry weight per plant of 11.55% and 6.45% respectively during the 2016 season. In a study conducted by Abdel Latef et al. [36] to assess the potential of two seaweed extracts *Sargassum muticum* and *Jania rubens*, a brown and a red macroalgae respectively, in improving growth and mitigating salinity in chickpea plants, the principal component analysis (PCA) showed that growth parameters are closely linked to the photosynthetic pigments. However, Spann and Little [37] demonstrated that the increase in shoot length of sweet orange nursery trees treated by a seaweed extract based on *Ascophyllum nodosum* during drought conditions could not be linked to photosynthesis. Moreover, no evident effect of the treatment during optimal conditions on citrus trees growth was recorded.

A greenhouse experiment was conducted to assess the effect of an *Ecklonia maxima* seaweed extract on growth and physiology of *Brassica rapa* L. under different nutrient concentrations (full strength (EC = 2.0 dS.m^−1^), half strength (1.0 dS.m^−1^) and tap water (0.5 dS.m^−1^)) [38]. Net photosynthesis was enhanced significantly in comparison to the control. The leaf area, shoot dry weight, and the marketable yield were significantly increased by 13.98%, 16.7% and 13.56% respectively. The treatment enhanced the leaf content in phosphorus and sodium, however, it had no effect on other macronutrients. The pigments concentration in *Brassica rapa* leaves were more pronounced in treated plants as the SPAD index was higher than the control by 14.7%. Concerning the leaf quality, the hydrophilic antioxidant activity (HAA) was higher than the control by 38.1%. However, ascorbic acid content and total phenols decreased, whereas a *Macrocystis pyrifera* seaweed extract significantly increased the antioxidant capacity, total phenols, and vitamin C content of cucumber fruits in comparison to Steiner solution treatment [39]. Moreover, the application of a commercial seaweed extract enhanced total soluble solids (TSS), fructose, and sucrose of strawberries which are compounds linked to fruit taste [40]. Furthermore, the application of *Ascophyllum Nodosum* seaweed extract at a rate of 1.5 kg.ha^−1^ significantly enhanced total anthocyanins, total phenols, and berry skin dry matter of grapevines. The veraison stage took place earlier than the control for Pinot Noir cultivar. And berries soluble sugars were enhanced by 30% and 22% for Pinot Noir and Cabernet Franc cultivars, respectively [41]. Interestingly, different seaweed extracts were applied during postharvest storage (Medium processing) of Fuji apple to investigate their effect on product quality. Results showed that *Codium tomentosum* seaweed extract decreased the browning index (BI) in homogenized and sliced apple by 33% and 26%, respectively, in comparison to the control and inhibited POD (peroxidase) and PPO (polyphenol oxidase) enzymes activity, which are synthetized during stress subjected to fruits as pathogens by means of scavenging and browning [42].

## 4. Biostimulant Properties of Seaweed Extract Are Emphasized under Abiotic Stress

Drought, salinity, extreme temperatures, and nutrients deficiencies are examples of the main factors inducing abiotic stresses, thus negatively affecting crop productivity. SE benefit is often associated to plant abiotic stress tolerance, although those positive impacts are still highly dependent on several aspects, including the targeted crop as well as the pedoclimatic conditions. Abiotic stress involves hyperosmotic and ionic imbalances leading to an oxidative stress resulting from the over production of reactive oxygen species (ROS) and the antioxidant defense mechanisms [43]. ROS production is the result of the reduction or activation of O_2_ implying the formation of singlet oxygen (^1^O_2_), superoxide (O_2_^−^), hydrogen peroxide (H_2_O_2_) and hydroxyl radical $\left(HO^{-}\right)$ [44]. ROS are known to impair biomolecules like DNA, proteins, and lipids [45]. To alleviate such deleterious effects, plant resort to several natural defensive mechanisms. More precisely, ROS are scavenged thanks to a system of enzymatic and non-enzymatic antioxidants such superoxide dismutase (SOD), guaiacol peroxidase (GPX), catalase (CAT), ascorbate peroxidase (APX), dehydroascorbate reductase (DHAR), glutathione reductase (GR), monodehydroascorbate reductase (MDHAR), ascorbic acid (AsA), phenolic compounds, glutathione and, tocopherol, etc. [45,46,47,48]. Latest studies conducted to investigate the effect of seaweed extracts to mitigate abiotic stress showed prominent results (Figure 3). For instance, Zou et al. [49] investigated the potential of polysaccharides extracted from *Lessonia nigrescens*, a brown macroalgae, to enhance salt stress adaptability of wheat seedlings. Polysaccharides enhanced significantly shoot and root lengths and dry and fresh matters of wheat under stress. They attenuated the oxidative damage of plants subjected to salinity by decreasing the relative electric leakage (REL) and malondialdehyde (MDA) content which are two parameters to assess membrane permeability and lipid peroxidation, respectively, and by increasing the antioxidant activity of SOD, POD, and CAT enzymes involved in ROS species scavenging. NaCl stress did not decrease the chlorophyll content in plants in comparison with the control, however, it was increased significantly when the polysaccharides were applied to stressed plants. Moreover, polysaccharides treatments maintained the osmotic status of stressed wheat seedlings by increasing their sugars content and proline and regulating the Na^+^/K^+^ ratio. Similarly, Liu et al. [50] studied the effect of polysaccharides from *Grateloupia filicina* in alleviating salt stress on rice during seed germination stage, and showed that they had stimulated rice seed development subjected to salinity. Moreover, an *Ascophyllum nodosum* extract decreased H_2_O_2_ content of Salam turfgrass plants, and increased CAT and APX activity and its ascorbate content in comparison with the control [51]. On a related note, Patel et al. [52] assessed the potential of *Kappaphycus alvarezii* sap (K sap) on alleviating salt and drought stress from three durum wheat varieties during vegetative and reproductive stages. K sap enhanced morphological parameters of durum wheat (shoot and root length and weight) in comparison to stressed plants. K sap increased total chlorophyll, carotenoids, and tissue water content; and reduced electrolyte leakage and lipid peroxidation (MDA). Under stress, K sap reduced the ionic imbalance by decreasing the Na^+^/K^+^ ratio and increasing calcium content, and the accumulation of osmoprotectants, proline, total protein, and amino acids. In vivo determination of ROS species showed less impact on plants treated with K sap under stress, however, no difference was noticed between the control and the treated plants under optimal conditions. The antioxidant potential was enhanced through the increase of the phenolic content and the non-enzymatic antioxidants. Additionally, Abscisic acid (ABA) and zeatin hormones increased both under stress and optimal conditions when K sap was applied. In a recent work, Sharma et al. [53] reported an increase of wheat biomass and yield by 57% and 70% respectively when a *Gracilaria dura* extract was applied to plants subjected to drought stress. According to the authors, the seaweed extract participated in facilitating the mechanisms involved in water saving strategies. Per contra, Trivedi et al. [28] investigated the effect of a seaweed extract based on *Kappaphycus alvarezii* on alleviating water deficit stress in maize and concluded that even if the level of some antioxidants was enhanced like APX, the yield was not improved significantly.

Seaweed extracts might be another solution to deal with low temperatures limiting the growth and development of plants. Algafect, a commercial seaweed extract based on *Ascophyllum nodosum*, *Fucus* spp. and *Laminaria* spp., reduced leaf necrosis and enhanced root length density of maize plants subjected to low root zone temperatures (12–14 °C) during two weeks [54].

Interestingly, seaweed extracts could be a potential technology to attenuate abiotic stress linked to nutrients deficiencies. Macroalgae extracts based on *Ascophyllum nodosum* succeeded in enhancing biomass fresh weight of lettuce [55], and shoot, leaf area and length and the sum of branches of almonds when plants were subjected to potassium deficiency [56]. Moreover, Carrasco-Gil et al. [57] experienced the effect of four commercial seaweed extracts based on *Ascophyllum nodosum* and *Durvillea potatorum* to alleviate iron deficiency on tomato plants. Some of the treatments were able to increase SOD and CAT enzymes in root and leaf. However, these extracts had no effect whatsoever on Fe uptake and acquisition mechanisms.

Priming is a promising technic in conveying to plant adaptability to stress [58]. It consists on keeping germination metabolism processes activated through the control of the hydric status of the seed [59,60]. Seed priming using seaweed extracts might be another approach to mitigate abiotic stress subjected to plants during germination and early growth stages. For instance, the application of a commercial seaweed extract (Kelpack) through priming, enhanced the germination rate of *Ceratotheca triloba* seeds under low temperatures fluctuating between 10 °C and 15 °C, and under low osmotic potential (−0.15 MPa) in comparison with the control [61]. Still, the treatment had no significant effect on germination rate when plants were subjected to different concentrations of NaCl (from 5 mmol.L^−1^ to 50 mmol.L^−1^). However, priming radish seeds subjected to 150 mM and 200 mM NaCl with *Codium taylorii* or *Pterocladia capillacea* extracts induced molecular changes in the DNA patterns of the plant by the disappearance of some bands and the appearance of other ones [62].

Overall, although several studies strongly demonstrated the ability of SE to alleviate abiotic stress, those effects are highly dependent on the algae species and the extraction method. Furthermore, according to our literature analysis, it seems that the plant species is also an important aspect to take into consideration as the physiological response of plants under a given environmental stress could drastically change from species to species or even among different lines of the same species.

## 5. Effect of Seaweed Extract on the Beneficial Microbiome of Plants: A Vital Component that Is Often Disregarded

Most of studies investigating SE mechanism of action focus solely on plant physiological changes, their plausible effects on the soil compartment is scarily disused despite its great importance, which can be attributed to SE mode of application that is often through foliar spray. For instance, SE could enhance PGP (plant growth promotion) traits of rhizopheric microbes. In an attempt to understand how seaweed extracts act on rhizosphere nutrients availability, several studies were conducted to elucidate their interaction with rhizosphere microorganisms and enzyme activity (Figure 2). For example, soil application of a commercial seaweed extract based on alginate on Chrysanthemum plants increased significantly rhizospheric-available P by 49% in comparison with the control. However, the soil microbial community was not affected [63]. In the same context, Renaut et al. [64] investigated the effect of a commercial seaweed extract based on *Ascophyllum Nodosum* on the soil and root microbiome diversity of pepper plants. Results revealed that the treatment significantly altered bacterial a-diversity in soil samples. Nevertheless, its effect was not clear in fungi a-diversity in soil and bacteria and fungi a-diversity in roots. In a recent study dedicated to assessing the quality of different soil samples degraded by erosion using biological indicators, a seaweed extract biostimulant based on *Macrocystis pyrifera* (8 mL) enhanced the hydrogenase activity (mg TPF/10 g) of the soil sample (10 g) having the lowest activity by 32% in comparison with the control [65]. Furthermore, a seaweed extract biostimulant based on *Lessonia nigrescens* and *Lessonia flavicans* applied at 40 g∙kg^−1^ of replant soil on *Malus hupehensis Rehd.* seedlings increased significantly the soil activity of invertase, urease, proteinase, and phosphatase enzymes in comparison with the control. Using the T-RFLP (terminal restriction fragment length polymorphism) analysis, they showed that the soil fungal community had been altered after the application of the seaweed extract at a ratio of 40 g∙Kg^−1^ [66]. Interestingly, invertase participates in catalyzing the hydrolysis of sucrose resulting in the formation of fructose and glucose [67]. Urease hydrolyses urea to form ammonia [68]. Protease hydrolyses proteins derived from soil organic matter and plays an important role in the nitrogen cycle [69]. Additionally, phosphatase plays a key role in altering organic phosphate from its complex or unavailable forms to assimilable ones [70]. On another note, seaweeds fertilizer based on *Lessonia nigrescens* and *Lessonia flavicans* greatly improved the number of beneficial microbes and fungi/bacteria ratio of *Malus hupehensis Rehd* rhizosphere, through establishing a moist environment favoring targeted development of beneficial microbes [71]. In the same manner biolog analysis of microbial metabolic activity of strawberry soil, as well as functional diversity, colony counts and soil respiration, increased in response to *Ascophyllum nodosum* treatment under field and greenhouse conditions [72]. Surprisingly, although those effects are of major importance they are still scarcely investigated and throughout studies using metagenomics tools are needed to clearly reveal the degree of microbial community discrepancy induced by SE application.

## 6. Seaweed Extract as a Potential Solution to Enhance Nutrient Use Efficiency

Since the green evolution, plant nutrient use efficiency (NUE) is drastically shrinking [73]. Consequently, enhancing NUE using sustainable technologies is one of the greatest challenges facing agriculture. In this context, many experiments were conducted to investigate the effect of seaweed extracts on plant nutrient uptake (Table 1). In a study conducted to evaluate the effect of six commercial seaweed extracts from *Laminaria spp* and *Ascophyllum nodosum* on maize used in a concentration of 0.5 mL/L during 48 h, leaf analyses showed that the ability of plants to absorb Ca, Mg, S, Fe, Cu, Mn, Mo, Zn, and B was enhanced significantly in comparison with the control [19]. Moreover, the application of a seaweed extract based on *Ascophyllum nodosum* on three clones of poplar plants enhanced significantly the K leaf content of an ‘Okanese’ clone under greenhouse conditions and the N stem content of ‘Okanese’ and ‘Walker’ clones under field conditions [74]. In another experiment devoted to evaluate the effect of *Ecklonia maxima* extract on growth and physiology of *Brassica rapa* L. under nutrient stress conditions, Di Stasio et al. [38] reported an increase of P and K leaf compositions of 30.5% and 20.7%, respectively. Furthermore, the authors recommended the use of a half-strength nutrient solution within their experimental conditions as it gives the same outputs in comparison with the full-strength one. Interestingly, Saa et al. [56] used rubidium as a tracer to assess almond K uptake when treated with different kinds of biostimulants, and showed that a seaweed extract from *Ascophyllum nodosum* fortified with amino acids and sugars enhanced significantly the leaf rubidium concentration of plants growing in a nutrient medium supplied with 5 µg∙g^−1^ of K, suggesting that this extract could be involved in increasing almond K uptake. Additionsally, Layek et al. [75] assessed the performances of two seaweed extracts based on *Kappaphycus alvarezii* and *Gracilaria edulis* in ameliorating productivity and quality of rice supplied with 100% recommended dose of fertilizers. Results showed that high concentrations (10% and 15%) of both extracts enhanced significantly the rice N and P uptake, but not K.

This enhancement in plant nutrient uptake using seaweed extracts could be attributed to their effect on the up regulation of some genes encoding root nutrients transporters. For instance, Jannin et al. [76] showed that the significant amelioration of *Brassica napus* N and S uptake treated with a seaweed extract based on *Ascophyllum nodosum* was associated to an over-expression of (BnNRT1.1; BnNRT2.1) and (BnSultr4.1; BnSultr4.2) which are genes encoding root transporters related to N and S uptake respectively. In the same context, three bioactive substances (251,104, NA9158, and EXT116) extracted from Laminariales, fucales, and ulvales were applied to the root of grapevine to investigate their effect on nutrient uptake and plant growth. Findings showed that EXT116 was the most effective in enhancing NH4^+^ and K^+^ influx in the roots part 0.8 and 1.7 mm from the root apex [77].

**Table 1 plants-09-00359-t001:** Effect of seaweed extracts on plant nutrient uptake and translocation.

Plant	Seaweed Species	Extraction Technic	Elemental Composition	Experiment Conditions	Mode of Application	Findings	Source
Maize	*Kappaphycus alvarezii (Ka)/Gracilaria edulis (Ge)* (Applied separately)	Liquid filtrate from fresh seaweed	K^+^ 33,654; 682.1, P^3+^ 17.45; not detected, Ca^2+^ 321; 352, Mg^2+^ 112; 311 for Ka and Ge respectively.	Field experiment	Foliar spray	Enhanced N, P and K uptake (grain + stover) for both extracts	[78]
Oilseed Rape	*Ecklonia maxima*	Cold cell burst	N 3.6 g.kg^−1^, P 8.2 g.kg^−1^, K 7.2 g.kg^−1^, Ca 0.8 g.kg^−1^, Mg 0.2 g.kg^−1^, Fe 13.6 mg.kg^−1^, Mn 8.4 mg.kg^−1^, B 0.24 mg.kg^−1^, Zn 4.2 mg.kg^−1^, and Cu 0.2 mg.kg^−1^	Pot experiment	Root application	Enhanced Leaf P and K concentration.	[38]
Tomato	*Ecklonia maxima*	Cold cell burst	N 3.6 g.kg^−1^, P 8.2 g.kg^−1^, K 7.2 g.kg^−1^, Ca 0.8 g.kg^−1^, Mg 0.2 g.kg^−1^, Fe 13.6 mg.kg^−1^, Mn 8.4 mg.kg^−1^, B 0.24 mg.kg^−1^, Zn 4.2 mg.kg^−1^, and Cu 0.2 mg.kg^−1^	In soil under greenhouse	Foliar spray	Enhanced Fruit Ca concentration.	[79]
Tomato	*Ascophyllum nodosum*	Not mentioned	Fe 39.9 µg.mL^−1^, Mn 20.9 µg.mL^−1^, Cu 3.0 µg.mL^−1^, Zn 4.1 µg.mL^−1^	Pots under growth chamber	Not mentioned	Enhanced concentration of Mn, Cu, and Zn in root and leaf.	[57]
Oilseed Rape	*Ascophyllum nodosum*	Acid extraction	Ca 0.854, Cu 0.009, Fe 0.030, K 6.630, Mg 0.919, N not detected, Na 3.102, P 0.116, S 2.660, Si 0.027, Zn 0.002% of dry weight	Hydroponic under greenhouse	In nutrient solution	Increased relative Mn, Cu, and Mg concentration in whole plant.	[80]
Wheat	*Ascophyllum nodosum*	Acid extraction	K 4442, P 78, S 1782, N 0 (concentration of 67 g DW dissolved in 1 L of water).	Pot experiment	Foliar spray	Enhanced Grain K	[81]
Oilseed Rape	*Ascophyllum nodosum*	Acid extraction	K 4442, P 78, S 1782, N 0 (concentration of 67 g DW dissolved in 1 L of water).	Pots under greenhouse conditions	In nutrient solution	Stimulation of root and shoot N and S.	[76]
Soybean	*Kappaphycus alvarezii*	Liquid filtrate from fresh seaweed	N 0.03%, P 33.99 mg.L^−1^, K 1.97%, S 0.06%, Ca 460.11 mg.L^−1^, Mg 581.2 mg.L^−1^, Na 0.51%, Cu 0.3 mg.L^−1^, Fe 10.59 mg.L^−1^, Mn 2.5 mg.L^−1^, Zn 0.62 mg.L^−1^	Soil, field experiment	Foliar spray	Enhanced N, P, K, S grain uptake and N, P straw uptake.	[82]

## 7. Seaweed Extract: A Complex Mixture with Multiple Mechanisms of Action

Natural extracts originate from treatment of vegetal raw materials and agricultural wastes, through chemical, physical or enzymatic process, thus the composition of the final product often encompass a broad spectrum of bioactive compounds that can theoretically induce multiple beneficial effects throughout plant development. Seaweed extracts (SE) are perfect examples of such rich composition. Additionally to minerals and polysaccharides, SE may also contain, depending on processing methods, phytohormones, cytokinins, vitamins, polyphenols, antimicrobial agents, and several other compounds of agronomical value [83,84,85,86,87]. Consequently, unraveling possible mechanism of action is often delicate and involve multi-disciplinary approaches through making use of sophisticated technics as to grasp the specific effect of SE at both the metabolic and genetic level (Figure 4 and Figure 5) (Table 2).

### 7.1. Mechanism of Action under Abiotic Stress

In an attempt to study the effect of monthly application of both seaweed extract and humic acid on growth of creeping bentgrass, Zhang et al. [88] reported an increased superoxide dismutase (SOD) activity (47% and 181%) which was accompanied by improvement of photosynthesis, resulting in a better bentgrass quality. Although, no clear evidence was given to explain how those substances my impact SOD activity, it was hypothesized that such phenomenon could be attributed to suitable hormonal and osmolyte content. Similar observations were noted by Elansary et al. [51] when investigating effects of foliar application of *Ascophyllum nodosum* extract on a *Paspalum vaginatum* cultivar, during prolonged irrigation intervals and under saline conditions. Those authors noted an improvement of lipid peroxidation by means of linoleic acid and DPPH (2,2′-dipheny-1-picrylhydrazyl) assays, and enhanced antioxidant defense mechanism underlined by significant escalation of SOD, CAT (catalase) and APX (ascorbate peroxidase) activities, leading to ROS (H_2_O_2_) depletion in SE treated plants. Furthermore, drought and salinity tolerance were imputed to cumulative effects, peculiarly a better photochemical activity attributed to mineral composition of SE and growth regulators like cytokinins and abscisic acid, enhanced root extension and modulation of root architecture, greater buildup of nonstructural carbohydrates, thus improving energy storage, metabolism and osmotic adjustment, and enhanced proline accumulation. Interestingly, even without SE application, those mechanisms are typical responses of plants subjected to drought or salinity constraints [89,90] and, manifestly, it seems that SE intervene on the whole defensive and adaptive system of plants in stressful conditions. This is in accordance with works of Shukla et al. [91] who reported specific amplification at the gene level of soybean natural defensive system under drought, following application of *Ascophyllum nodosum* extract. In fact, quantitative RT–PCR analysis of genes directly related to stress revealed that for instance, GmCYP707A1a and GmCYP707A3b, two genes involved in regulating ABA biosynthesis during dehydration and hydration cycles, were overexpressed due to SE application; and such response was not recorded before stress, as gene expression remained unchanged in both control and SE treated plants. On another note, Jithesh et al. [92] experienced the effect of an *Ascophyllum nodosum* extract on *Arabidopsis thaliana* under salinity conditions using transcriptomics and physiological analyses. The plants treated with an ethyl acetate sub-fraction of *Ascophyllum nodosum* showed an up-regulation of 184 and 257 genes at the first and fifth days, respectively, and a down regulation of 91 and 262 genes at day 1 and 5, respectively. Genes related to the abiotic stress group constituted 2.2% of the overall up regulated genes at day 1 and increased to 6% at day 5. Moreover, the application of an *Ascophyllum nodosum* extract (7 mL∙L^−1^) on asparagus on a weekly basis under salinity conditions (2000 and 4000 ppm NaCl) enhanced significantly the up-regulation of ANN1, ANN2, and PIP1, an aquaporin and water management related genes, and P5CS1 and CHS, a two biologically-active molecule metabolism-related gene. However, redox related genes APX1 and GPX3 followed the same up regulation in comparison with the control. Moreover, the experiment revealed the potential of the treatment in alleviating salinity stress by the enhancement of phenols, proline, and antioxidant activities [35].

Under drought conditions, a *Gracilaria dura* extract altered the abscisic acid homeostasis of wheat by the up regulation of some NCED (9-cis-epoxycarotenoid dioxygenase) genes like TaNCED3.1 and TaNCED3.2. This was associated with the significant enhancement of ABA content in comparison with the control [53]. Similarly, and to assess the Arabidopsis stomatal closure caused by drought stress in both control and treated plants by an *Ascophyllum nodosum* extract, the expression of genes related to the biosynthesis of ABA was followed. Results showed an up regulation of NCED3, a gene implicated in the ABA biosynthetic pathway, RAB18 and RD29A, two sensitive genes to ABA starting from the 3rd day following dehydration for the untreated plants. However, this up regulation was slightly stable to moderate for the treated ones, suggesting that the seaweed extract might be responsible in hampering the function of the mechanisms related to stress [93]. Furthermore, transcript expression of an abiotic stress-responsive transcription factor TaWRKY10, the ROS scavenging genes TdCAT, TdSOD, and a stress signaling cascade gene WCK-1 of wheat plants under drought conditions were upregulated after the application of *Kappaphycus alvarezii* sap (K sap). The later, was the only gene that was upregulated in comparison to the control when the treatment was applied alone. Consequently, the authors suggested that the induction of phytohormones after the application of K sap could be the cause of the genes and transcription factors upregulation under stress conditions [52]. Additionally, tomato plants were treated with seaweed extracts based on *Ascophyllum nodosum* (AN1, AN2, and AN3) before and after applying a drought stress by withdrawing water during a period of 7 days. The AN1 treatment was the most effective in enhancing the relative water content (RWC), fresh weight biomass, soluble sugars and proline contents, and decreasing the MDA content during the T1 date equivalent to the end of the stress period. Meanwhile, both drought and AN1 increased the up regulation of the Tas14 gene, which is a stress protective protein [94]. In a recent study, an *Ascophyllum nodosum* extract applied to tomato and sweet pepper plants enhanced significantly the up regulation of gene transcripts Ga_2_Ox, IAA, and IPT implicated in the biosynthesis of gibberellin, auxin, and cytokinin, respectively [95]. The authors hypothesized that this over expression might be linked to the enhancement of plant growth and development. Furthermore, Billard et al. [80] used a seaweed extract based on the same species *Ascophyllum nodosum* on oilseed rape, and showed that the treatment enhanced significantly the expression of COPT2, a gene coding for a Cu transporter, BnSULTR1.1 and BnSULTR1.2, genes coding for sulfate transporters, and BnNRT1.1 and BnNRT2.1, genes coding for nitrate transporters. Interestingly, the relative Cu concentration in the whole treated plants was significantly higher than the control.

**Table 2 plants-09-00359-t002:** Effect of seaweed extracts on plants’ gene expression.

Plant	Seaweed Species	Findings	Sources
Wheat	*Lessonia nigrescens*	Downregulation of TaHKT2; 1, a transporter implied in Na^+^ uptake from the soil, and the upregulation of TaSOS1 and TaNHX2 antiporters functioning in the Na^+^ exclusion to vacuoles.	[49]
Soybean	*Ascophyllum nodosum*	Over expression of stress responsive genes GmRD22, GmDREB, GmFIB1a, GmERD1, GmBIPD during drought stress and GmPIP1b during the recovery stage.	[91]
Arabidopsis	*Ascophyllum nodosum*	ANE A and B dysregulated the expression of 4.47% and 0.87% of the transcriptome respectively.	[96]
Oilseed Rape	*Ascophyllum nodosum*	Over expression of (BnNRT1.1; BnNRT2.1) and (BnSultr4.1; BnSultr4.2) which are genes encoding N and S root transporters respectively.	[76]
Arabidopsis	*Ascophyllum nodosum*	Up regulation of cold related genes CB73, RD29A, and COR15A.	[97]

### 7.2. Mechanism of Action under Biotic Stress

SE potency become truly conspicuous under peculiar pressures including biotic constraints. In fact, disease control traits are often associated to SE application [98]. For instance, chloroform and benzene extract of brown seaweed *Padina pavonica* demonstrated nymphycidal activity and the ability to significantly reduce or increase the nymphal development period, by interfering with *Dysdersus cingulatus* (Cotton pest) physiology. Such effect was attributed to anti-juvenile hormonal content of SE, obstructing pest development at germ bund or blastokineis stages [99]. Furthermore, *Ulva lactuca* powder applied as soil treatment at the rate of 5 g/kg significantly diminished root knot nematode infections in banana plants, through reduction of number of galls (76%) and final population of nematodes, which was directly correlated with SE phenolic contents [100]. SE were also evaluated for their antifungal properties and solvent extract of some macroalgae specifies isolated from Brazilian Manguinhos beach (i.e., *H. musciformis*, *O. secundiramea*, *P. capillacea*, *L. dendroidea*, and *P. gymnospora*) successfully inhibited *Colletotrichum gloeosporioides*, a fungal species inducing anthracnose in papaya and banana plants, which severely hamper fruits postharvest. GC-MS analysis of extracts showed that the highest antifungal effect was correlated with presence of halogenated terpenes, fatty acids (hexadecanoic and octadecanoic acids), and quercetin [101]. Additionally, to aforementioned antimicrobial properties of their bioactive molecules, SE may also play an eliciting role through triggering specific responses against pathogens. Indeed Cluzet et al. [102] demonstrated that an *Ulva* spp. mixture extract, sprayed at only 5 µg/mL, protected *Medigaco truncatula* against *Colletotrichum trifolii*. Data analysis following RNA extraction of plant materials and targeting genes related to defense mechanisms revealed that genes encoding critical enzymes involved in phytoalexin and phenylpropanoid biosynthesis and key pathogenesis analogous genes, were distinctly up-regulated (i.e., phenylalanine ammonia-lyase, chalcone synthase, isoflavone reductase, chitinase, etc.). Authors noted that such up-regulation was not achieved at the expanse of primary metabolism as SE extract prevented down-regulation of genes related to carbon and nitrogen metabolism, which is often a systematic response of plants facing pathogenic coercion [103]. Likewise, in vitro assays showed that SE extract did not impair *C. trifolii* development, thereupon suggesting that the prompted protection could solely be due to eliciting attributes. 

## 8. Novel Approach in SE Development: The Promise of the Algae Bio-Refinery Concept

Even if SE are currently the dominant category of the plant biostimulants segment with a market predicted to reach a value of 894 million Euro in 2022 [11]. Several challenges need to be addressed as to gain economic viability and to achieve true suitability. First and foremost, the industry is highly dependent on biomass availability. Indeed, due to national regulation of coastal area, exploitation of macroalgae for industrial applications is exclusively restricted to highly available or invasive species. Thus, the important biodiversity remains untapped, not mentioning that such approach could ultimately leads to over-exploitation. Furthermore, according to our market survey (involving SE manufacturers and distributers), the price of those products can vary between $30 and $70 per acre, which remains high and clearly not affordable for all categories of farmers. Finally, the scientific approach employed in SE manufacturing is fundamentally flawed, as the currently used methodology in macroalgae harvesting could ultimately leads to product inconsistency due to seasonal and environmental variations. Consequently, the question arising is what is the best strategy for implementing a solid industry allowing the development of consistent, efficient and economically viable products? Within the recent technological developments, the bio-refinery concept seems to be highly suitable model for optimized macroalgae exploitation. The bio-refinery concept originates from the petroleum industry. It revolves around an integrated process involving several operations allowing the full exploitation of raw material and the generation of high value co-products; hence, economic viability, energy consumption, as well as waste management aspects are further enhanced [104]. Bio-refinery concept have been proposed for fuel production from microalgae [105,106,107]. Likewise, bio-refinery based on seaweed have also been discussed by several authors. For instance, Zollman et al. [108] proposed a concept for a seaweed bio-refinery based on tailored cascade process (horizontal and vertical), that exploit the properties of each fraction as to generate various products, including plant biostimulants, human and animal food, and biofuel. Those authors also insisted on the necessity of offshore cultivation through using methodologies that does not require nutrient supplementation, which should alleviate labor and energy inputs. Such suggestion could prove to be one of the turning point of SE manufacturing. Indeed, in contract to classical approaches, which are mainly based on algae harvesting form natural ecosystems, cultivation would allow the growth of algae in a more controlled environment. Such initiative would ideally offer a double benefit, namely, less variability in terms of chemical and biochemical composition, and more importantly, the cultivation conditions could be theoretically designed as to enhance the yield of targeted compounds with biostimulant features. On another note, Álvarez-Viñas et al. [109] presented in their review a successful bio-refinery example focusing on red seaweed processing. Those authors highlighted several scenarios for the implementation of a cascading process based on a very interesting classification emphasizing the content of some targeted bio-compounds, namely the agarophytes, the carragenophytes, and the porphyranophyte. Consequently, numerous processing pathways could be envisioned. Interestingly, this could be a potent method for the establishment of SE biostimulant specialized bio-refinery. In a more specific manner, Ingle et al. [110] suggested a model designed for low-income countries through exploiting the red algae *Kappaphycus alvarezii*. This model based on a flux balance analysis allowed the identification and the implementation of the most efficient fermentation strategies for economically viable ethanol production. Based on their predictions, such model could lead to the production of biofertilizers (67%), carrageenan (12%), and ethanol (77.6 g.kg^−1^). Although such concept does not integrate SE biostimulants, it provides valuable data with regards to the potential and attractiveness of such approach, as the significant yield of high value co-products could create opportunities as to further enhance SE production process without affecting the economical balance.

Overall, the current postulate suggests that the SE industry should change their “focus on a sole product” mindset and the bio-refinery concept would be the adequate strategy to move forward. The inconsistency of plant response due mainly to composition variability of seaweed would be then solved; the economic impediment associated to the development of efficient extraction processes would be less important. Finally, the implementation of cultivation techniques would allow less dependency on raw material availability, thus aligning with future regulation of natural resources exploitation within coastal areas and providing opportunities for more investors.

## 9. Seaweed Extract Biostimulants: Legal Acts

The European Union is becoming the first legislative organization to recognize plant biostimulants as a separate group of agricultural inputs [111]. The Fertilising Products Regulation (FPR) (EU) 2019/1009 was published recently, and defined a plant biostimulant as “a product the function of which is to stimulate plant nutrition processes independently of the product’s nutrient content with the sole aim of improving one or more of the following characteristics of the plant or the plant rhizosphere: (a) nutrient use efficiency, (b) tolerance to abiotic stress, (c) quality traits, or (d) availability of confined nutrients in the soil or rhizosphere” [112]. To summarize, seaweed extracts were categorized within the non-microbial plant biostimulant group. The concentration of contaminants in the extract should not exceed some defined concentrations, e.g., 1.5; 120; 1 mg∙kg^−1^ dry matter are the thresholds for Cd, Pb, and Hg, respectively. Moreover, the concentration of Cu and Zinc have not to exceed 600 and 1500 mg/kg dry matter respectively. Similarly, the concentrations of *Salmonella* spp., *Escherichia coli* or *Enterococcaceae* pathogens must be verified in the extract. A critical point outlined in the statute is that the biostimulant must have the effects claimed on the product label for the plants listed on that. This could represent a challenge for the manufacturers as the above-mentioned literature shows a variability of outputs after the application of the seaweed extracts. In this context, a recent paper was published by Ricci et al. [113] to suggest some guidelines to adopt when trying to prove the plant biostimulants claims. In overall, they proposed that the claims have to be in agreement with the European Union definition of plant biostimulants, using existing literature and accessible data, and setting up experimental trials with respect to numerous points as the experimental design, the treatments application conditions, and the mode of assessment.

## 10. Concluding Remarks and Future Prospects

It is obvious that SE biostimulants are continuously gaining more importance within the agricultural sector. As demonstrated by the current review, this advancement was achieved through recent scientific knowledge describing their mechanism of action and affirming, to some extent, industrial claims. Moreover, the recent regulation of plant biostimulants within the EU will bring new insights to the expansion of seaweed extracts in the recent future. However, considering the present findings discussed in this paper, gaps are still lingering. Consequently, more efforts are needed to bring SE extracts to the next level, and the following suggestions would be their turning point:✓**Gaining the farmer trust**: Meeting farmer’s requirements is vital in the success of any given agri-input, which can only be achieved through adopting a farmer proximity approaches focusing on field demonstration, education as well as generating data specifically tailored for the end-customer. Furthermore, future research dealing with SE must focus on providing a manual of good farmer practices consisting of the best method of application, rates, frequencies, and time of application, etc. This would be the missing piece in the way of exploring the high potential of SE and harmonizing their final effect under different environments.✓**The transition from a niche to a broader market**: SE and more generally plant biostimulant are still viewed as niche market, as they exclusively target horticulture and specialty crops. This can be explained by several factors, including the lack of consistency and rigorous regulation, their application method, and the farmers’ impressions. This latter is of high importance as SE biostimulants are still exhibited as an eco-friendly alternative for classical fertilization, which is erroneous and detrimental for the industry. Instead, SE should be aligned with mineral fertilization, thus making the most of a wider market through focusing on enhancing fertilizer eco-efficiency.✓**Breaking down SE mode of action**: Biostimulants as they are formulated currently should be considered as unique cases requiring typical approaches. Indeed, additionally to their complex composition which makes it difficult to precisely pinpoint bioactive molecules inducing a specific effect, they are lacking broad-spectrum properties resulting in inconsistency, not always fully explained as the plant/soil domain is an intricate ecosystem governed by a string of variables including pedoclimatic conditions, crop types and soil microbiome distinctive features. It is mandatory to breakdown those complex mixtures to single molecule to ascertain occurring synergisms and antagonisms. The challenge may seem at first sight insurmountable, as the number of data to analyze is simply overwhelming. That said research is undoubtedly making progress through exploitation of novel high-throughput approaches (i.e., high-throughput sequencing and phenotyping, metabolomics, etc.).✓**Enhancing SE agri-economic efficiency**: Several aspects need to be addressed to improve the agronomic efficiency of SE. Such issues could be mainly attributed to the peculiar nature of seaweed biomass, as its composition could radically change depending on the collection timing and the overall environmental conditions. Moreover, as described in our review the extraction process is still not fully optimized. Rising to the challenge will require the establishment of novel integrated approach as to fully control the manufacturing steps. In this regard, algal biorefinery concept integrating controlled algae cultivation is highly promising. This could resolve both the inconsistency in composition problem, as well as the product pricing. Ultimately, given the positive impact of SE beneficial plant microbes, furthering synergies between SE and microbial biostimulants could be the key to the development of next-generation plant biostimulants.

## Figures and Tables

**Figure 1 plants-09-00359-f001:**
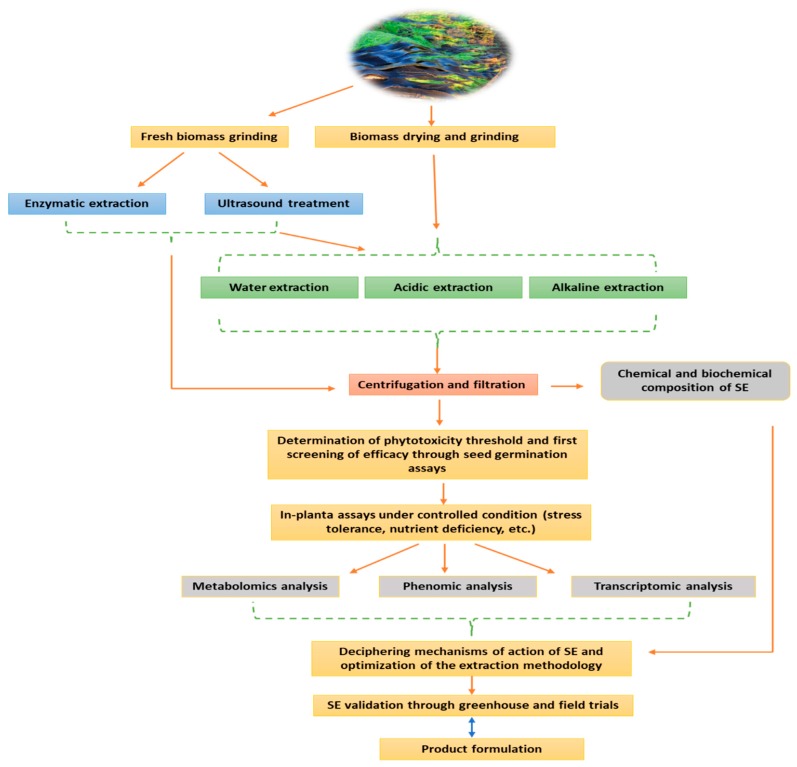
Proposed flowchart of seaweed extraction process from collected algal biomasses until product validation. The flowchart is based on routinely used extraction methodologies as well as high-throughput approaches for the validation of the product efficacy.

**Figure 2 plants-09-00359-f002:**
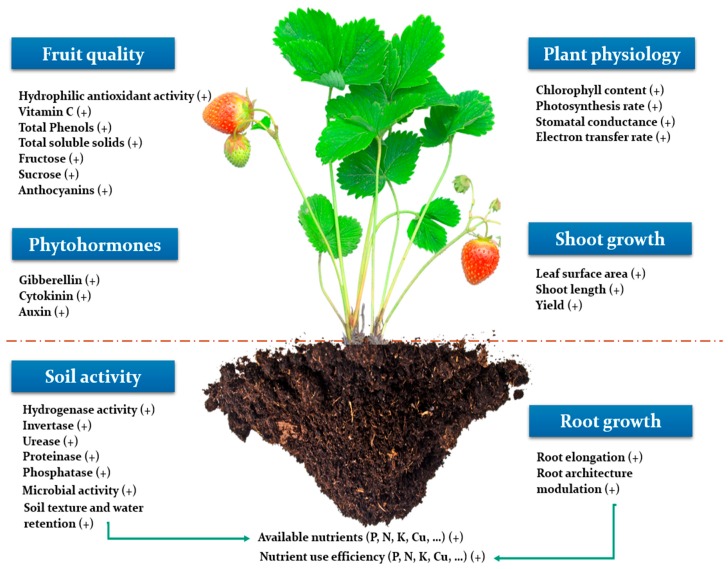
Conceptual illustration highlighting the positive impact of seaweed extracts on the whole soil–plant system. Such effects encompass improving fruit quality, and plant phytohormone content, increasing soil enzymatic activity, improving the rooting system and the overall physiological features of plants.

**Figure 3 plants-09-00359-f003:**
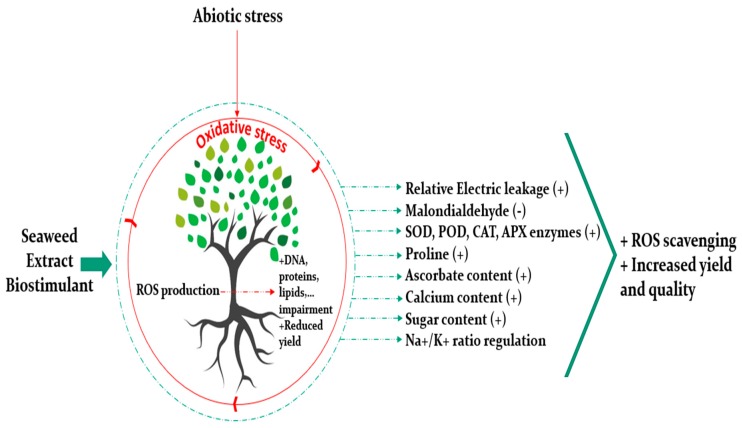
Beneficial effect of seaweed extracts under abiotic stress: Seaweed extracts play an important role in reactive oxygen species scavenging (ROS) through triggering several mechanisms involving stimulation of antioxidants and inhibition of lipid peroxidation. Superoxide dismutase (SOD), peroxidase (POD), catalase (CAT), ascorbate peroxidase (APX).

**Figure 4 plants-09-00359-f004:**
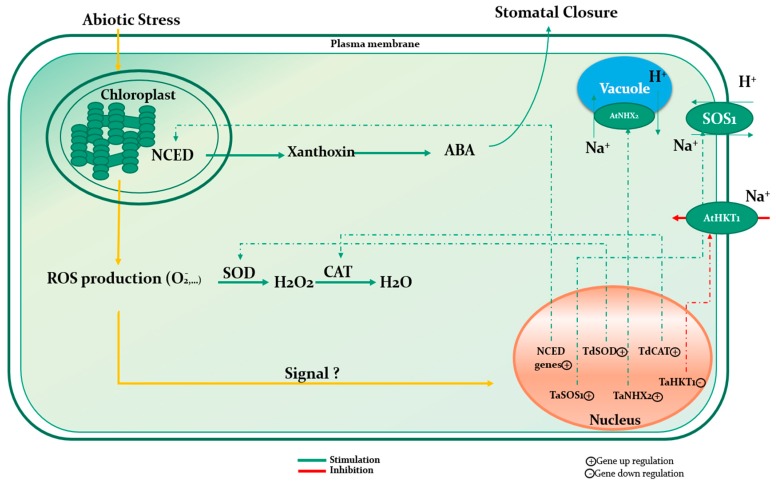
Conceptual illustration highlighting the effect of seaweed extract on the down and up-regulation of some key genes in response to abiotic stress: Seaweed extract application to plants subjected to abiotic stress alter the regulation of ROS scavenging related genes (TdSOD and TdCAT), NCED (9-cis-epoxycarotenoid dioxygenase) related genes, Na^+^ transporter (TaHKT1) and antiporters (TaSOS1 and TaNHX2) genes. Superoxide dismutase (SOD), catalase (CAT), abscisic acid (ABA).

**Figure 5 plants-09-00359-f005:**
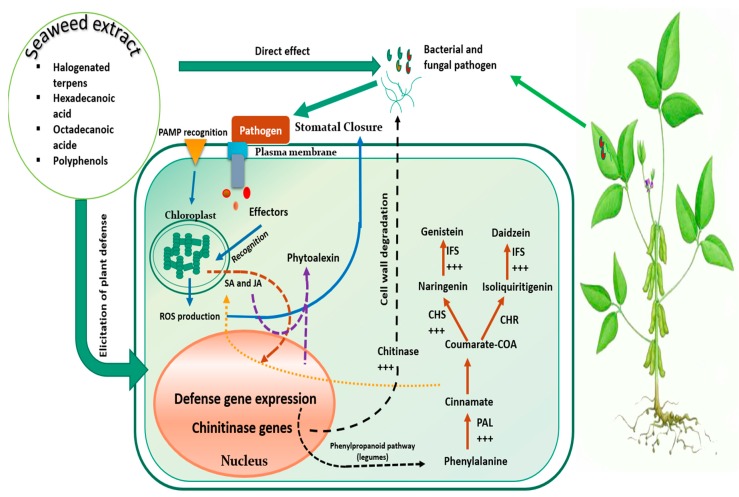
Conceptual illustration highlighting plausible mechanisms of action of seaweed extract with respect to fungal and bacterial disease control: seaweed extract may affect directly pathogen agents or indirectly through eliciting the plant defense machinery Such elicitation occurs via the up regulation of several defense genes, thus positively affecting specific metabolic pathways, namely the phenylpropanoid pathway, which induce the biosynthesis of several secondary metabolites involved in disease suppression. JA: Jasmonic acid, SA: Salycilic acid, PAL: Phenylalanine ammonia-lyase, CHS: Chalcone synthase, CHR: Chalcone reductase, IFS: Isoflavone synthase, IFR: Isoflavone reductase.

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
