# Peer review of "Trends in Seaweed Extract Based Biostimulants: Manufacturing Process and Beneficial Effect on Soil-Plant Systems"

_plants, 2020, doi:10.3390/plants9030359_

Round 1

Reviewer 1 Report

An attempt has been made in the study to describe the efficacy and applications of seaweed extract-based biostimulants in agricultural production. The topic has been addressed comprehensively, starting with the effects of the analyzed compounds on nutrient uptake by plants, and ending with their influence on plant microbiome. The manufacturing process of the biostimulants has also been described, including the relevant regulations. This review article summarized the often fragmentary information available in the existing literature. The manuscript is well written and well organized, and it can be very interesting to readers. The review has been carefully prepared, and enriched with figure drawings and diagrams. The issues listed below should be addressed during the revision process, although they detract nothing from the scientific merit of the study.

The reference entries in the Reference section should be numbered because otherwise it is difficult to check whether the in-text citations are correct. Lines 45, 49, 125, 396, 409, 483, 524, 534 and 540 – replace the authors’ names with numbered citations, as elsewhere in the manuscript. In in-text citations such as [27] [28] and others, the authors’ names should be given, e.g. Trivedi et al. [27]… – please correct here and throughout the manuscript. Figure 1 is too large for the margins of the document – its size should be proportionally reduced (maintaining the adequate resolution) or the figure should be modified graphically by moving its elements. The same applies to Figures 2, 3 and 5. Line 237 – the formulas of chemical compounds, singlet oxygen and superoxide, should be correctly written, and a missing space should be inserted; line 238 – remove the unnecessary space. Table 1 is too large – font size and column width should be reduced. In units of measure, replace a period with a half-high dot, e.g. mg∙kg-1. Correct also the units in lines 191, 215, 328, 332, 352 (missing spaces), 364, 421, 473, 565, 567. Please keep Figure 4 and its caption on one page. The symbols of cations in Figure 4 should have the plus sign “+” in superscript. The same applies to line 377 and Table 2. Section 9 heading could be changed to “Seaweed extract biostimulants: legal acts”.

The following corrections should also be made:

– line 203 – in the unit of measure, the number “-1” should be in superscript,

– line 261 – it should be K sap, not Ka sap,

– lines 291 and 493 – a period is missing at the end of the sentence,

– lines 291, 306 and 461 – replace the capital letter “S” with the small letter “s” in the word “Seaweed”,

– lines 296 and 297 – correct the unit of measure, because the concentration of a substance in a solution is expressed as the number of moles of solute per unit volume or mass of solution,

– line 332 – correct the unit of measure, insert a space,

– lines 372, 373 and 642 – remove the unnecessary space,

– line 379 – the section heading should be followed by at least two lines of text belonging to this section,

– line 401 – correct the formula of the chemical compound,

– line 454 – replace capital letters with small letters in the phrase “Oilseed Rape”,

– line 465 – remove the unnecessary colon in Table 2 caption,

– Table 2 – remove unnecessary spaces and insert missing spaces,

– References – in the titles of publications, the Latin names of species should be italicized,

– lines 648,786, 880 – bibliographic data are missing,

– lines 662, 665, 681, 721, 745, 751, 759, 765, 783, 840, 849, 859, 863, 865, 898, 904 – the numbers of manuscripts or page numbers are missing,

– lines 678, 687, 708, 714, 783, 826, 836, 861 – journal titles are capitalized,

– lines 716 and 810 – the same reference, differing only in the year of publication, is given twice,

– lines 739 and 741 – the numbers in chemical formulas should be in subscript,

– line 764 – remove {},

– lines 816 and 869 – the same reference entry,

– line 849 – replace the small letter in the journal title with a capital letter,

– lines 886, 887, 892 – correct the notation of authors’ names.

Author Response

Dear reviewer

We appreciate the interest that you have taken in our manuscript and the constructive criticism you have given. We have addressed the major concerns of the manuscript. Changes in the text are visible through using microsoft word fuction”track changes”.

All the references are now numbered in the text In citation such us [27] and [28] authors name have been given The size of some figures has been adjusted accordingly and the formula of chemical compounds was corrected Table size have been adjusted and units in line 191, 215, 328, 332, 352 364, 421, 473, 565, 567 have been corrected. Section 9 title have changed to “Seaweed extract biostimulants: legal acts” Some missing page number of manuscript (line 662, 687…) in the list of references have been added Modification in line 203, 261, 291, 493, 379, 401… have been made according to reviewer suggestions. Please let us know if you have any further remarks or suggestions.

Thank you again for time spent reviewing our manuscript

With best regards,

Karim LYAMLOULI  

Reviewer 2 Report

Dear Authors, you should address my recommendations highlighted 
across the text.

Author Response

Dear reviewer

We appreciate the interest that you have taken in our manuscript and the constructive criticism you have given. We have addressed the major concerns of the manuscript. Changes in the text are visible through using microsoft word fuction”track changes”.

All the references are now numbered in the text All errors related to punctuation have been corrected Sentence in the introduction section (line 50, 51 and 52) is not incomplete as a comma separate “industrial sectors” and “once again” not a point. That being said we have changed the sentence as to be more comprehensive. In line 87 the “etc” have been removed as we have stated the most common bioactives compounds in seaweed. Capital letters are now used for the each section title. In citation such us [27] and [28] authors name have been given The size of some figures has been adjusted accordingly and the formula of chemical compounds was corrected Table size have been adjusted and units in line 191, 215, 328, 332, 352 364, 421, 473, 565, 567 have been corrected. Section 9 title have changed to “Seaweed extract biostimulants: legal acts” Some missing page number of manuscript (line 662, 687…) in the list of references have been added Modification in line 203, 261, 291, 493, 379, 401… have been made according to reviewer suggestions.

Please let us know if you have any further remarks or suggestions.

Thank you again for time spent reviewing our manuscript

With best regards,

Karim LYAMLOULI